# Validation of the Generative Acts Scale-Chinese Version (GAS-C) among Middle-Aged and Older Adults as Grandparents in Mainland China

**DOI:** 10.3390/ijerph18199950

**Published:** 2021-09-22

**Authors:** Haoyi Guo, Steven Sek-yum Ngai

**Affiliations:** Department of Social Work, Chinese University of Hong Kong, Shatin, Hong Kong, China; syngai@cuhk.edu.hk

**Keywords:** generative acts, grandparents, middle-aged, older adults, measurement invariance, China

## Abstract

The current study examined the psychometric properties of the 20-item Generative Acts Scale-Chinese version (GAS-C) among middle-aged and older adults as grandparents in mainland China. A total of 1013 grandparents (mean age = 58.32 years; 71.9% female) of children from 12 kindergartens were recruited using multistage cluster random sampling. A four-factor pattern of domestic, agentic, communal, and civic generative acts were identified by exploratory factor analysis and further verified by confirmatory factor analysis (CFA). Subsequently, multigroup CFA was performed to test the measurement invariance across gender, age group and *hukou* status at the configural, metric, and scalar levels. The Cronbach’s alpha value of the total (0.923) and subscales (range from 0.897 to 0.953) was satisfactory, indicating high internal consistency. Additionally, the significant gender differences in the domestic (male = 3.565, female = 3.718, *p* < 0.05), communal (male = 2.786, female = 2.591, *p* < 0.01), and civic subscales (male = 2.112, female = 1.864, *p* < 0.001) and the significant correlations between the GAS-C total scale and subscales with caregiving intensity (*r* = 0.433, *p* < 0.01), positive affect (*r* = 0.397, *p* < 0.01) and life satisfaction (*r* = 0.328, *p* < 0.01), supported concurrent validity. Overall, this study addressed the knowledge gap by validating a reliable and valid instrument to measure grandparents’ generative acts in mainland China, contributing to generativity studies cross-culturally in research and practice.

## 1. Introduction

First introduced by Erikson [1], generativity is concerned with establishing, nurturing and contributing to the younger generations, through actions such as childrearing, teaching, mentoring, volunteering, and community activities [2,3,4]. Although originally applied to mid-life adults, the concept of generativity has extended to older adults who are increasingly involved as grandparent caregivers worldwide [5] (pp. 1–18). Caregiving to grandchildren has been viewed as an expression of generativity in later life that tends to yield tremendous health benefits to grandparent caregivers [6]. Western literature has extensively documented the positive role of generativity regarding grandparent caregivers’ personal growth, satisfaction, and psychological well-being [3,4,7]. In addition, generativity has been found to be negatively associated with depressive symptoms among Korean grandmothers [8]. Influenced by traditional Confucianism that prescribes intergenerational ties, generative acts such as grandparent caregiving are prevalent in the mainland. Surprisingly, generativity studies among Chinese grandparents are rare. Lacking a reliable scale of generative acts with adequate psychometric properties in the mainland China context, could be one of the major reasons. To address the measurement gap, it has become essential to validate the Chinese version of the Generative Acts Scale with a sample of Chinese grandparents in the mainland.

### 1.1. Generative Acts among Chinese Grandparents

With extended longevity and improved health conditions, older adults spend more years with grandchildren and shoulder increased childrearing responsibilities as grandparent caregivers [9] (pp. 2–3). Caring for grandchildren as a cultural norm is prevalent in rural and urban China [10,11]. Evidence from the China Health and Retirement Longitudinal Study (CHARLS) suggests that 58% of Chinese grandparents are caregivers for their grandchildren [12]. Rural grandparents are estimated to take primary caring responsibilities for approximately 29 million left-behind grandchildren due to parental out-migration [13]. In urban China, grandparent caregivers also provide childcare extensively on a daily basis to facilitate parents’ labor participation, given the limited supply of formal childcare services [14,15]. Besides, urban China hosts an emerging group of migrant grandparent caregivers from rural homes, who join adult children to care for grandchildren in urban destinations [16]. According to the latest statistics from the National Health Commission of the People’s Republic of China, there are around eight million grandparent caregivers with migration backgrounds [17]. Moreover, with a growing number of Chinese graduate students and research fellows pursuing further education overseas, it is common for them to invite parents to be carers for young children in a foreign country [18]. Likewise, Chinese American grandparents have been found highly involved in caregiving for grandchildren, reporting an average of 11.96 caring hours per week from the Population Study of Chinese Elderly in Chicago (PINE) [19]. In sum, Chinese grandparents widely serve as caregivers for grandchildren as a way of conducting generative acts.

### 1.2. Instruments to Measure Generative Acts

Considering the enormous number of Chinese grandparent caregivers in the mainland and overseas, it is imperative to understand the caregiving behaviors and related practices of generative acts, using a culturally valid and reliable instrument. However, measurements of generativity remain scarce among grandparents. The most commonly used scale is the 20-item Loyola Generativity Scale (LGS) [2]. It focuses on participants’ perceptions of their generative concerns influenced by social norms and individual needs [3]. An example statement would be, “I feel as though I have made a difference to many people.” Notably, the Loyola Generativity Scale neglects to measure the real behaviors arising from generative concerns as caregivers. Except for LGS, the Generative Behavior Checklist (GBC) was devised to capture the frequency of performing specific generative actions [2,20] (pp. 7–43). However, the 50-item length of the second scale could be challenging for older adults in self-administrating questionnaires. Moreover, it does not contain items exclusively on caregiving, such as serving meals for grandchildren.

The latest scale targeting generative acts was developed by Cheng [21] with a sample of older adults in Hong Kong. The Generative Acts Scale (GAS) consists of 20 items with subscales of civic acts (four items) and nonspecific acts (sixteen items). An example statement would be, “I take care of children and grandchildren when they are ill.” Compared to the LGS and the GBC, the GAS has several advantages: (1) it has a reasonable number of items, (2) it describes exact generative behaviors and contains caregiving activities, (3) it covers features of generativity at the individual, family, and community levels, and (4) it is developed from a Chinese sample in Hong Kong with items culturally adapted to the Chinese population. Yet, the GAS’s Chinese version has not been tested in the mainland China context. Due to Hong Kong’s unique historical and social context, it is thus necessary to assess the GAS-C’s psychometric properties with grandparents in mainland China before applying it to other relevant studies.

### 1.3. Aim of This Study

Living in a highly Westernized metropolitan city, Hong Kong’s older adults may have a different level of civic participation, a key component of generative acts, compared with their counterparts in mainland China [22]. Meanwhile, the age of becoming a grandparent in the mainland, which can be as early as one’s 40s, is much earlier than what is typical in industrialized Hong Kong [23]. The probability of being a grandparent is over 80% for Chinese on the mainland when they reach 55 years old [24]. In other words, the age group of Chinese grandparents could be either middle-aged or older adults. Moreover, the language of the original scale is Cantonese among Hong Kong samples, which is different from the official Mandarin language used in mainland China. As mentioned, the distinctive social contexts, demographic differences, and language barriers justify a separate test of the GAS-C in mainland samples.

Apart from the above factors, the study aims to examine the applicability of the GAS-C for measuring grandparents’ generative acts in the mainland China context. We will test for: (1) a conceptually meaningful factor structure of the GAS-C using exploratory factor analysis (EFA) and confirmation factor analysis (CFA); (2) measurement invariance of the scale across gender, age groups, and hukou status using multigroup CFA; (3) internal consistency reliability; and (4) concurrent validity of the GAS-C by analyzing the gender differences, the relationship between caregiving intensity and generative acts, and correlations between scores on the GAS-C total scale and subscales with those on the positive affect and life satisfaction measures. Furthermore, findings from the National Survey of Midlife Development in the United States (MIDUS) reveal that caregivers may experience greater self-perceived generativity than non-caregivers [25]. As mentioned, grandparent caregiving is one of the specific performances of generative acts, and we thus propose that grandparent caregivers with a higher caregiving intensity will report a higher level of generative acts as measured by the GAS-C.

## 2. Materials and Methods

### 2.1. Participants and Procedure

The data were collected in Cixi City of Eastern China from September 2020 to November 2020 using structured questionnaires with participants chosen on a multistage cluster random sampling basis. In China, as in many other countries, informal caregivers such as grandparent caregivers are not documented officially, but children from three to six years old go to officially registered kindergartens. Enrollment lists of all kindergartens in Cixi City were obtained from the education bureau. First, four districts were randomly selected in Cixi City, then three kindergartens in each district were randomly selected, coming to a total of 12 kindergartens. Each child’s surviving grandparent (only one was needed to answer the questionnaire if there was more than one) in the 12 kindergartens constituted the sample. The inclusion criteria were (1) to be a grandparent of a child aged 3–6 years, and (2) co-residing with the grandchild or living nearby. Grandparents having more than one grandchild in the kindergarten were counted once only. The final sample was 1013 grandparents. Their average age was 58.32 years (SD = 7.48), ranging from 40 to 93 years old. Among the participants, 58.3% belonged to the middle-aged group (*n* = 591) of 40 to 59 years, and 41.7% belonged to the older adults group (*n* = 422) of 60 years and above. Grandmothers accounted for 71.9% of the total sample (*n* = 728), with 28.1% being grandfathers (*n* = 285). Of the grandparents, 761 (75.1%) co-resided with grandchildren, and 252 grandparents (24.9%) lived nearby. Around half of the participants possessed a local hukou (*n* = 505), and another half were migrants without a local hukou status (*n* = 508). The vast majority (91.6%) had a junior school or below educational level, one out of four (26.7%) had a chronic disease, and one out of ten (9.6%) were divorced or widowed. Most of the participants (79%) had retired or engaged in part-time jobs, and a dominant majority (89.7%) had a monthly income of RMB 5000 or below (around USD 768). This study was approved by the Survey and Behavioral Research Ethics Committee at the institution where the authors were affiliated. Informed consent was obtained before the survey.

### 2.2. Measurement

Generative acts were measured by the 20-item Generative Acts Scale developed by Cheng [21] from a sample of community-dwelling older adults in Hong Kong. Participants were asked to indicate the frequency of doing generative acts in the past two months on a five-point Likert scale ranging from “1 = Almost none” to “5 = Very often”. The original scale consists of generative acts–civic (Items 3, 4, 10, 19) and generative acts–nonspecific. A sample item of the civic subscale would be, “Participate in volunteer work and continue to serve the community.” The remaining 16 items were not divided into detailed subscales. Sample statements would be, “I take care of the grandchildren when their parents are not available,” and “I pass on my skills and talents to the next generation.” The scale’s Chinese translation (Cantonese version) was obtained from Cheng [21] and adapted to the Mandarin version before use. A higher score indicated a higher level of generative acts. The Cronbach’s alpha for this scale is 0.923 in the present study.

Caregiving intensity was measured by the hours spent each day caring for the grandchild(ren). Considering the time requirement for a full-time job is eight hours per day, we set half of that amount (four hours) as the threshold of high intensity—with reference to a study in Europe [26]—and a quarter of eight hours (two hours) as the threshold of medium intensity. Caregiving intensity was a categorical variable, with 1 = 0 h, 2 = 1–2 h (low intensity), 3 = 3–4 h (medium intensity), and 3 = 5 h or above (high intensity).

Positive affect was measured with the Positive Affect Scale among grandparents in this study [27]. The original positive affect scale had ten items, and we used an internationally reliable short form of five items [28]. Respondents indicated the extent of their feelings of positive affect (excited, enthusiastic, inspired, alert, and determined) of the past week on a five-point Likert scale, from “1 = Very slightly or not at all” to “5 = Very much”. Higher positive affect scores represent having full energy, high concentration, and pleasant engagement [27]. The Cronbach’s α for PA was 0.905.

Life satisfaction was measured using the five-item Satisfaction with Life Scale (SWLS) on a five-point Likert scale ranging from “1 = Strongly disagree” to “5 = Strongly agree”. Participants responded to statements such as “In most ways, my life is close to my ideal” and “I am satisfied with my life” based on their feelings of the past week [27]. A mean score was calculated to indicate participants’ life satisfaction. A higher score suggested higher life satisfaction. Cronbach’s alpha for this scale is 0.906 in the present study.

### 2.3. Data Analyses

First, descriptive statistics were compiled to describe the social demographic characteristics of the participants. Then an exploratory factor analysis (EFA) on subsample A (random split-half, *n* = 506) was performed to determine the factor patterns, with principal axis factoring as the extraction method and Promax as the rotation method. Items with factor loadings less than 0.30 and absolute loadings above 0.32 on two or more factors were considered for deletion [29]. After obtaining the model structures from EFA, a confirmatory factor analysis (CFA) was conducted to verify the initial subscales of the Generative Acts Scale. We adopted the model fit criteria proposed by Brown [30] (pp. 67–74) and Hu and Bentler [31]: the comparative fit index (CFI) exceeded 0.90, and the robust root mean square error of approximation (RMSEA) and standardized root mean square residual (SRMR) were below 0.08.

Subsequently, multigroup CFA was performed to assess the scale’s construct validity between males and females, middle-aged and older adults, and locals and migrants. Moreover, we tested measurement invariance (configural, metric, and scalar) across groups of gender, age group, and hukou status by using the chi-square difference test, changes in the comparative fit index (ΔCFI), and changes in the root mean square error of approximation (ΔRMSEA). Nonsignificant results of the chi-square difference test [32], |ΔCFI| < 0.01 [33], and |ΔRMSEA| < 0.015 [34] would indicate invariance. Cronbach’s alpha value checked the internal consistency for the GAS-C and each subscale. Since there were multiple factors as dependent variables, multivariate analysis of variance (MANOVA) was used to examine the mean scores of all subscales across gender groups and the relationships between the GAS-C and caregiving intensity. In addition, bivariate Pearson correlations (two-tailed) of the total scale and subscale scores with life satisfaction and positive affect were obtained to verify concurrent validity. Data analysis was conducted with SPSS 24.0 (released by IBM Corp. in 2016, Armonk, NY, USA) and Mplus 8.0 (released by Muthen & Muthen in 2017, Los Angeles, CA, USA).

## 3. Results

### 3.1. Exploratory Factor Analysis

Since the original GAS did not specify subscales, an EFA with principal axis factoring extraction was performed to identify the scale’s factor structure. Assuming the subscales were correlated, the Promax rotation method was used. Given the sufficient sample size, a random split-half subsample (*n* = 506) was selected [35]. The Kaiser–Meyer–Olkin value was 0.916, and Bartlett’s test of sphericity was nonsignificant (*p* < 0.001), suggesting the sample met the factor analysis criteria [36]. As shown in Table 1, a rotated factor loading matrix presented a four-factor solution from EFA, explaining 76.95% of the total variance. Each item had a single dominant factor loading. The first factor contained four items (Items 1, 2, 7, 8) about the caregiving activities and was named “domestic generative acts”. The second factor included five items (Items 5, 12, 13, 14, 15) related to the extension of the self [37] (pp. 7–18), which we thus termed “agentic generative acts”. Six items (Items 6, 9, 11, 16, 17, 18) about mentoring and caring for youth without biological bonds [38] were considered “communal generative acts”. The remaining items (Items 3, 4, 10, 19) were initially termed “civic” by its developer [21]. In this study, Item 20 was also loaded onto the fourth factor. Considering the meaning of Item 20 (“doing something that benefits others”) was relevant to the civic factor, we thus branded the fourth factor as “civic generative acts” with five items (Items 3, 4, 10, 19, 20). With no items requiring removal based on EFA results, the four-factor solution, blending theoretical classifications from Kotre [37], McAdams, Hart and Maruna [20] and Villar [39], was tentatively accepted before further investigation.

### 3.2. Confirmatory Factor Analysis

A CFA on subsample B (*n* = 507) was subsequently performed to confirm the four-factor structure of GAS-C obtained from EFA. Given that all items were normally distributed with absolute values and skewness and kurtosis below one [40], the default maximum likelihood (ML) estimation method was adopted in Mplus. The original model demonstrated satisfactory model fit (χ^2^ = 559.093, *df* = 164, *p* < 0.001, CFI = 0.955, TLI = 0.948, RMSEA = 0.069, and SRMR = 0.043). As shown in Figure 1, all standardized factor loadings exceeded 0.70 and were statistically significant at 0.001. In addition, the four subscales significantly and positively correlated with each other. In particular, the domestic subscale and agentic subscale were positively and strongly correlated (0.65), both of which were at the individual/family level; in contrast, the positive and strong correlation of the communal subscale and civic subscale (0.65) suggested generative acts at the societal level [39]. Overall, the CFA results yielded sufficient construct validity of the four-factor model of the GAS-C.

### 3.3. Measurement Invariance

Next, we tested the measurement invariance by gender (males and females), age group (middle-aged and older adults), and hukou status (locals and migrants), using the total sample (*n* = 1013). As presented in Table 2, satisfactory results were obtained from the following subsamples: males (*n* = 285, χ^2^ = 411.739, *df* = 164, *p* < 0.001, CFI = 0.945, RMSEA = 0.077, and SRMR = 0.047); females (*n* = 728, χ^2^ = 767.230, *df* = 164, *p* < 0.001, CFI = 0.951, RMSEA = 0.071, and SRMR = 0.043); middle-aged (*n* = 591, χ^2^ = 694.756, *df* = 164, *p* < 0.001, CFI = 0.946, RMSEA = 0.074, and SRMR = 0.043); older adults (*n* = 422, χ^2^ = 557.346, *df* = 164, *p* < 0.001, CFI = 0.948, RMSEA = 0.075, and SRMR = 0.044); locals (*n* = 508, χ^2^ = 595.253, *df* = 164, *p* < 0.001, CFI = 0.949, RMSEA = 0.072, and SRMR = 0.044); and migrants (*n* = 505, χ^2^ = 580.713, *df* = 164, *p* < 0.001, CFI = 0.953, RMSEA = 0.071, and SRMR = 0.043).

To further confirm the construct validity of the GAS-C, three levels of invariance models across gender, age group and hukou status were carried out as documented in Table 3. First, we tested the configural invariance by constraining the factorial structure to be the same between each subgroup. The goodness-of-fit of the configural models indicated that the four-factor structure of the GAS-C was equivalent across gender, age group and hukou status. Then, metric invariance was tested by constraining the factor loadings in the multigroup analysis. After comparing metric models with baseline models using the chi-square test, ΔCFI, and ΔRMSEA, the results showed that the factor loadings of gender subgroups (Δχ^2^ = 15.815, *p* = 0.466, ΔCFI = 0.000, ΔRMSEA = 0.002), age subgroups (Δχ^2^ = 22.957, *p* = 0.115, ΔCFI = 0.001, ΔRMSEA = 0.002) and hukou subgroups (Δχ^2^ = 55.207, *p* < 0.001, ΔCFI = 0.003, ΔRMSEA = 0.000) were invariant. We continued to test the scalar invariance by constraining the item intercepts to be equal across each subgroup. The changes of indices on gender (Δχ^2^ = 51.815, *p* = 0.015, ΔCFI = 0.001, ΔRMSEA = 0.003), age (Δχ^2^ = 42.143, *p* = 0.108, ΔCFI = 0.001, ΔRMSEA = 0.003), and hukou (Δχ^2^ = 141.356, *p* < 0.001, ΔCFI = 0.006, ΔRMSEA = 0.001) indicated that the intercepts of each item were also invariant across these groups. Though the chi-square value, which was sensitive to sample size, was significant in the hukou subgroups, changes in CFI and RMSEA were both lower than the thresholds, thus supporting the measurement invariance test.

### 3.4. Reliability Estimation

Having established the validity of the GAS-C, we further evaluated the reliability of the items, the scale, and the subscales. The item reliability was assessed using the corrected item-dimension correlations, and the internal consistency of the scale and subscales was estimated using Cronbach’s alpha value. According to Table 4, the item-dimension correlations of the domestic, agentic, communal, and civic subscales were 0.679–0.847, 0.742–0.828, 0.807–0.884, and 0.746–0.832, respectively, thus demonstrating a high degree of homogeneity within each subscale. The Cronbach’s alpha value of the whole scale was 0.923, while the subscale values were 0.897 for the domestic subscale, 0.909 for the agentic subscale, 0.953 for the communal subscale and 0.912 for the civic subscale. These results demonstrated that the GAS-C, together with all dimensions, was reliable.

### 3.5. Concurrent Validity

Table 5 presents the means, standard deviations, and multivariate tests of gender differences in the total scale and four subscales. Females scored significantly higher than the male group in domestic generative acts (male = 3.565, female = 3.718, *p* < 0.05), while females exhibited significantly lower scores in communal (male = 2.786, female = 2.591, *p* < 0.01) and civic (male = 2.112, female = 1.864, *p* < 0.001) generative acts. There was no gender difference in agentic generative acts (male = 3.988, female = 3.948, *p* > 0.05). Arguably, the relatively high score on the domestic subscale and relatively low scores on the communal and civic subscales of the female group probably counterbalanced the differences in the total mean score of the male group (male = 3.113, female = 3.030, *p* > 0.05). Despite that, significant gender differences showed in three out of four subscales in the current sample, namely, domestic, communal, and civic generative acts.

Additionally, the bivariate Pearson’s correlation test showed the GAS-C total scale significantly and positively correlated with caregiving intensity (*r* = 0.433, *p* < 0.01). MANOVA results further showed that generative acts were positively associated with caregiving intensity on the domestic subscale (F (3, 1009) = 210.526, *p* < 0.001), agentic subscale (F (3, 1009) = 60.689, *p* < 0.001), communal subscale (F (3, 1009) = 12.067, *p* < 0.001), and civic subscale (F (3, 1009) = 4.263, *p* < 0.01). As displayed in Figure 2, grandparents who had never cared for grandchildren (0 h in caregiving intensity) exhibited the lowest scores in all forms of generative acts. When caregiving intensity increased to 1–2 h per day, the chances that grandparent caregivers were conducting generative acts increased dramatically. Such an upward trend continued till the caregiving intensity reached 3–4 h and beyond 5 h per day, though the steepness of the slopes was slightly reduced. Moreover, the mean scores in domestic (3.675) and agentic (3.960) generative acts were much greater than communal (2.646) and civic (1.933) generative acts.

Next, we examined the correlations of the GAS-C total and subscale scores with positive affect and life satisfaction. As shown in Table 6, the GAS-C was significantly correlated with measures of positive affect (*r* = 0.397, *p* < 0.01) and life satisfaction (*r* = 0.328, *p* < 0.01). Apart from the total scale, all the subscales of the GAS-C were significantly and positively correlated (*p* < 0.01) with positive affect and life satisfaction. According to the thresholds advanced by Cohen [41], correlations above 0.30 suggested medium to great effect size; thus, the correlation coefficients supported the GAS-C’s concurrent validity.

## 4. Discussion

Overall, the findings provided partial support for the reliability and validity of the 20-item Generative Acts Scale-Chinese version (GAS-C) to apply among Chinese grandparents in the mainland. We extracted four factors from EFA and initial CFA results. Four subscales were thus labeled as the domestic subscale (four items), agentic subscale (five items), communal subscale (six items), and civic subscale (five items) based on the item content, the original author’s categorization [21], and previous research on generativity [20,37,39]. Factor loadings on each item of the latent sub-construct were between 0.78 and 0.93. The subsequent measurement invariance test by multigroup CFA revealed that the GAS-C had configural, metric, and scalar invariances across gender, age group, and hukou status, indicating adequate construct validity of the GAS-C. Thus, the measurement model’s factorial structure, factor loadings, and intercepts were equivalent between males and females, middle-aged and older adults, and locals and migrants. With Cronbach’s alpha coefficients of the subscales and the whole scale larger than 0.80, the GAS-C also demonstrated sufficient internal consistency.

Moreover, an examination of gender differences in four types of generative acts exhibits good concurrent validity of the GAS-C. Specifically, females (grandmothers in this sample) conducted significantly more domestic generative acts, such as preparing meals, caring for grandchildren, and doing housework. This finding echoes the view of gendered roles in informal caring, which argues that women assume more childrearing responsibilities than men [42]. On the other hand, males (grandfathers in this sample) were more involved in communal and civic generative acts than females. For example, males were more likely to counsel younger people or to participate in volunteer work. Similarly, Western studies of generativity have discovered that men score high in cultural, technical, and social generativity, for instance, in guiding young people and making positive changes in society [43]. We further noted that the average subtotal scores on domestic (3.675) and agentic (3.960) subscales are much higher than for communal (2.646) and civic (1.934) subscales, indicating that participants tend to be more involved at the individual/family level rather than the community/society level in conducting generative acts. Villar and Serrat [38] propose that the double-faceted nature of generative acts could link personal and societal development when proper interventions are implemented. By enhancing community facilities, organizing peer groups, or enacting age-friendly social policies, Chinese grandparents may be encouraged to perform more generative acts at the societal level, which, in turn, would benefit their own families or themselves. Moreover, our hypothesis that caregiving intensity was positively associated with generative acts was confirmed in the current study. Grandparents who spent more time caring for grandchildren were more likely to conduct generative acts of each type. In addition, the positive and significant correlations of the GAS-C with positive affect and life satisfaction signified the psychological benefits of being generative, which is in line with empirical findings from Western studies [7,44]. These consistent relationships provide additional evidence for the concurrent validity of the GAS-C. Based on the above analyses, we consider the GAS-C a valid and reliable instrument for measuring grandparents’ generative acts in mainland China.

Concerning the GAS-C’s factor structure, Cheng’s (2009) original work had two factors: civic (four items) and nonspecific (sixteen items). Our study enriches the literature by providing unique empirical findings on a distinct four-factor structure of the GAS-C: domestic (four items), agentic (five items), communal (six items), and civic (five items) subscales. Our study includes a larger sample (*n* = 1013), a wider age range (middle to old age), a more diverse household registration composition (locals and migrants), and a different sociocultural context (e.g., the practice of grandparent caregiving). This addition to both the scope and the specificity of the instrument illustrates the difference between the original Hong Kong version and the current mainland China questionnaire. Having realized that generative acts in later life often extend beyond the family, for instance, in the form of civic engagement [45] (pp. 227–263), Cheng (2009) grouped four items (Items 3, 4, 10, 19) regarding civic generative acts from the 188 participants (60 to 89 years) in his research. In comparison, our study detected two factors beyond the family: communal (Items 6, 9, 11, 16, 17, 18) and civic (Items 3, 4, 10, 19, 20) generative acts. Item 20, “Do something that benefits others,” also fell onto the civic factor based on statistical analysis, and we deem such a revision as reasonable considering its content. The communal factor clearly observed in our study was not captured in Cheng’s (2009) study. Regarding the sample’s age, the mean age in Cheng’s (2009) study was 73.0 years, nearly 15 years older than our sample’s mean of 58.3 years. Such a disparity may result in a decline of the physical capacity in carrying out generative acts in the community due to aging [46].

The two factors (domestic and agentic) on the family/individual level were also spotted in the current study, initially grouped as “nonspecific” by Cheng (2009). On the one hand, childrearing, traditionally viewed as a developmental task for the middle-aged, has been increasingly regarded as an expression of generativity for older adults who serve as grandparent caregivers [47]. On the other hand, being grandparent caregivers does not necessarily mean entering older adulthood. Like many developed countries, Hong Kong, as a highly industrialized and modernized society, has delayed marriages—a first marriage often occurring while in one’s 30s—in the past decades, resulting in a coincidence of grandparenthood and older adulthood [48] (pp. 421–437). Yet, in mainland China, the chances of first-time grandparents are 80% when a couple reaches 55 years, which means many middle-aged adults serve as grandparent caregivers there [24]. Mostly in their robustness and influenced by the traditional familism that emphasizes intergenerational solidarity [49], it is common for Chinese grandparents to co-reside with the adult children’s family, especially migrant grandparents who leave rural hometowns to provide grandchild care in urban destinations. In the current study, grandparents assume much of the domestic work and transmit their life experiences to their offspring, which turn up in domestic (Items 1, 2, 7, 8) and agentic (Items 5, 12, 13, 14, 15) subscales. These two factors were not specifically extracted from the Hong Kong sample, probably because of the differences in housework arrangement and caregiving practice. Since the 1980s, Hong Kong has introduced tens of thousands of domestic helpers from Southeast Asian countries; dual-career families widely resort to paid workers at home instead of relying solely on grandparents for domestic work and caregiving responsibilities [50] (pp. 91–112). In summary, social demographic and contextual differences could have led to different patterns of generative acts in the current sample that eventually resulted in the four distinct factors being observed in this study. In other words, our study has demonstrated the cultural applicability of the GAS-C by showing its sensitivity to the unique sociocultural context of mainland China, which is quite different from that of Hong Kong, even though both of them are influenced by Chinese culture and have Chinese as the major population group [51].

This study is subject to some limitations. First, since the research site is only one city in Eastern China, its generalizability to the other parts of China needs further evidence. Future studies may try to collect data in different regions of China with consideration of various backgrounds, such as economic development level, rurality level, and ethnicity. Second, due to the nature of informal care, such as grandparent caregiving, it is almost unfeasible to acquire a complete sampling frame, as grandparent caregivers are not registered officially. Thus, we chose registered kindergartens as the sampling frame to randomly select sample elements (children in kindergartens). Thirdly, the data was collected at one time, and future studies could consider additional intervals over a longer period to assess the test-retest reliability. Apart from these limitations, our study is the first measurement study of the Generative Acts Scale among Chinese grandparents ranging from middle to old age, which deepens the understanding of generative acts cross-culturally and expands the knowledge of generativity into the middle-aged and older adult population.

## 5. Conclusions

Our study has validated the 20-item Generative Acts Scale-Chinese version (GAS-C) with adequate psychometric properties among middle-aged to older adults as grandparents in mainland China. The unique four-factor pattern, domestic, agentic, communal, and civic subscales, extracted from the original two-factor scale, has demonstrated sufficient internal consistency reliability, construct validity, and concurrent validity, enabling cross-cultural comparison with generativity research in the Western context. Moreover, the GSA-C’s measurement invariance has been tested by gender, age group and hukou status, suggesting its applicability to a variety of populations (males and females, middle-aged and older adults, and locals and migrants). Overall, the findings fill in the knowledge gap of measuring generative acts in the mainland China context, and enrich the literature by specifying different types of generative acts, which would contribute to generativity research and practice in future.

## Figures and Tables

**Figure 1 ijerph-18-09950-f001:**
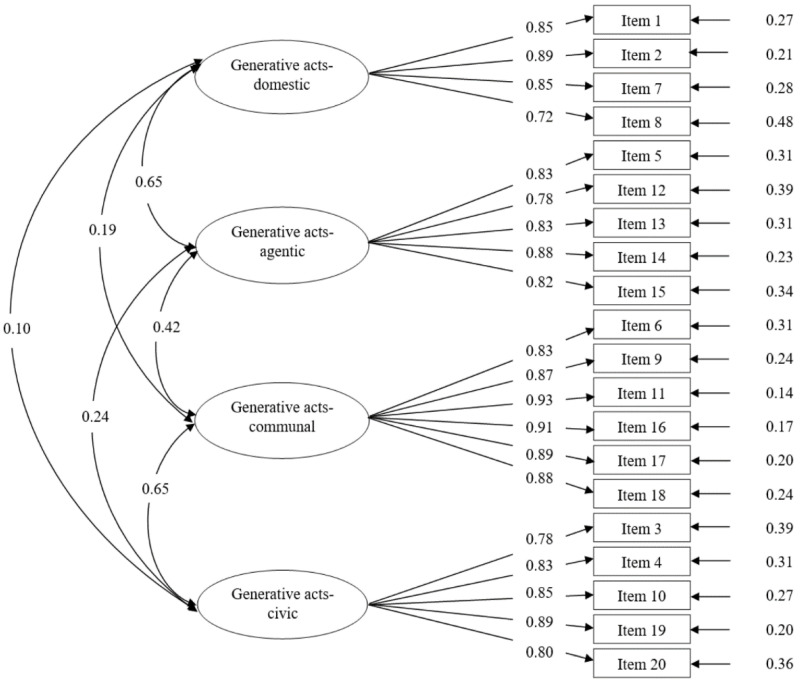
Validation of the factor structure with CFA (Subsample B, random half; *n* = 507). All coefficients in Figure 1 were standardized factor loadings statistically significant at *p* < 0.001 level.

**Figure 2 ijerph-18-09950-f002:**
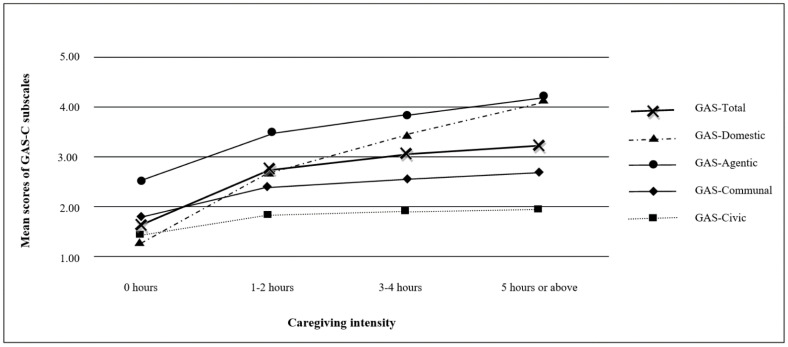
Mean scores of the GAS-C scale and subscales by caregiving intensity.

**Table 1 ijerph-18-09950-t001:** Rotated factor loadings matrix from EFA (Subsample A, random half; *n* = 506).

Items	F 1	F2	F 3	F 4
1. I take care of children and grandchildren when they are ill.	**0.837 ^1^**	0.029	−0.001	0.051
2. Taking care of my offspring’s daily life, including preparing meals.	**0.977**	−0.051	0.006	−0.027
7. Take care of the grandchildren when their parents are not available.	**0.792**	0.077	0.007	−0.029
8. Do housework for my children.	**0.676**	0.056	−0.008	0.023
5. Be a role model to the next generation.	0.252	**0.673**	−0.010	−0.016
12. Share my past experience, whether bitter or sweet, with the next generation.	0.055	**0.721**	0.009	−0.022
13. Teach the next generation not to spend money on unnecessary items.	−0.041	**0.869**	−0.057	−0.003
14. Teach the next generation to know right from wrong, and to observe rules and regulations.	−0.040	**0.943**	−0.041	0.001
15. Pass on my skills and talents to the next generation.	0.049	**0.686**	0.134	0.033
6. Learn new things so as to make myself useful to the younger generations.	0.024	−0.035	**0.806**	0.042
9. Teach the younger generations how to get along with others and handle various matters.	0.002	0.037	**0.878**	−0.033
11. Take initiative to comfort young people in distress.	0.012	−0.009	**0.925**	−0.043
16. Counsel younger people who are emotionally disturbed.	−0.020	−0.015	**0.935**	−0.042
17. Encourage the younger generations to learn new things and develop multiple interests.	−0.032	0.028	**0.870**	0.011
18. Teach younger generations to do voluntary work and to serve others.	0.030	−0.042	**0.802**	0.129
3. Participate in volunteer work and continue to serve the community.	0.068	−0.045	−0.053	**0.839**
4. Visit other people in need, like patients.	0.049	−0.092	−0.044	**0.887**
10. Participate in community educational activities.	−0.026	0.026	−0.013	**0.861**
19. Give a hand to needy people in the community.	−0.045	0.027	0.098	**0.808**
20. Do something that benefits others.	−0.067	0.129	0.120	**0.620**

^1^ Factor loadings larger than 0.30 for each item are in bold.

**Table 2 ijerph-18-09950-t002:** Structural validation categorized by gender, age, and hukou.

	Gender		Age		Hukou	
	Male (*n* = 285)	Female (*n* = 728)	Middle-Aged (*n* = 591)	Older Adults (*n* = 422)	Local (*n* = 508)	Migrant (*n* = 505)
Chi-square	411.739	767.230	694.756	557.346	595.253	580.713
Degrees of freedom	164	164	164	164	164	164
*p* value	<0.001	<0.001	<0.001	<0.001	<0.001	<0.001
CFI	0.945	0.951	0.946	0.948	0.949	0.953
RMSEA	0.077	0.071	0.074	0.075	0.072	0.071
SRMR	0.047	0.043	0.043	0.044	0.044	0.043

Note. CFI comparative fit index, RMSEA root mean square error of approximation, SRMR standardized root mean square residual.

**Table 3 ijerph-18-09950-t003:** Gender-, age-, and hukou-related measurement invariance of the GAS-C.

Model	χ^2^	*df*	CFI	RMSEA	Δχ^2^	Δ*df*	*p*	ΔCFI	ΔRMSEA
By gender									
Configural invariance	1208.969	328	0.949	0.073					
Metric invariance	1224.783	344	0.949	0.071	15.814	16	0.466	0.000	0.002
Scalar invariance	1260.643	360	0.948	0.070	51.674	32	0.015	0.001	0.003
By age									
Configural invariance	1252.102	328	0.947	0.075					
Metric invariance	1275.059	344	0.946	0.073	22.957	16	0.115	0.001	0.002
Scalar invariance	1294.245	360	0.946	0.072	42.143	32	0.108	0.001	0.003
By hukou									
Configural invariance	1175.966	328	0.951	0.071					
Metric invariance	1231.173	344	0.949	0.071	55.207	16	<0.001	0.002	0.000
Scalar invariance	1317.323	360	0.945	0.072	141.357	32	<0.001	0.006	0.001

Note. *χ^2^* Chi-square, *df* degree of freedom, CFI comparative fit index, RMSEA root mean square error of approximation, Δ*χ^2^* difference between models’ *χ*^2^, Δ*df* difference between models’ *df*, *p*: *p*-value, ΔCFI difference between models’ CFI, ΔRMSEA difference between models’ RMSEA.

**Table 4 ijerph-18-09950-t004:** Description and reliability estimation of the GAS-C items and subscales.

Subscales	Items	M	SD	Corrected Item-Dimension Correlation	Cronbach’s Alpha of Subscales
Generative acts-domestic	Item 1	3.578	1.147	0.794	0.897
	Item 2	3.698	1.215	0.847	
	Item 7	3.978	1.131	0.786	
	Item 8	3.445	1.329	0.679	
Generative acts-agentic	Item 5	4.019	1.006	0.781	0.909
	Item 12	3.769	1.102	0.742	
	Item 13	4.086	0.976	0.772	
	Item 14	4.186	0.946	0.828	
	Item 15	3.738	1.184	0.758	
Generative acts-communal	Item 6	2.499	1.046	0.807	0.953
	Item 9	2.688	1.086	0.854	
	Item 11	2.720	1.082	0.884	
	Item 16	2.601	1.094	0.876	
	Item 17	2.763	1.129	0.864	
	Item 18	2.604	1.159	0.846	
Generative acts-civic	Item 3	1.696	0.917	0.746	0.912
	Item 4	1.651	0.888	0.792	
	Item 10	2.015	1.072	0.812	
	Item 19	1.956	1.034	0.832	
	Item 20	2.350	1.068	0.715	

**Table 5 ijerph-18-09950-t005:** Mean comparisons of the total and subscales of the GSA-C by gender.

Variable	*n*	Mean	SD	Range	F (1, 1011)
Male (total)	285	3.113	0.714	1–5	2.950
Female (total)	728	3.030	0.678	1–5	
Domestic (subtotal)	1013	3.675	1.056	1–5	4.322 *
Male	285	3.565	1.036	1–5	
Female	728	3.718	1.061	1–5	
Agentic (subtotal)	1013	3.960	0.897	1–5	0.402
Male	285	3.988	0.817	1–5	
Female	728	3.948	0.926	1–5	
Communal (subtotal)	1013	2.646	0.990	1–5	8.001 **
Male	285	2.786	1.073	1–5	
Female	728	2.591	0.951	1–5	
Civic (subtotal)	1013	1.934	0.859	1–5	17.474 ***
Male	285	2.112	0.941	1–5	
Female	728	1.864	0.814	1–5	

Note. * *p* < 0.05, ** *p* < 0.01, *** *p* < 0.001.

**Table 6 ijerph-18-09950-t006:** Concurrent validity: Bivariate Pearson Correlations.

	M	*SD*	1	2	3	4	5	6	7
1. GAS-Total	3.053	0.689	1						
2. GAS-Domestic	3.675	1.056	0.707 **	1					
3. GAS-Agentic	3.960	0.897	0.780 **	0.626 **	1				
4. GAS-Communal	2.646	0.990	0.762 **	0.220 **	0.408 **	1			
5. GAS-Civic	1.933	0.859	0.647 **	0.133 **	0.219 **	0.594 **	1		
6. Positive affect	3.058	0.712	0.397 **	0.203 **	0.325 **	0.335 **	0.299 **	1	
7. Life satisfaction	3.448	0.915	0.328 **	0.156 **	0.298 **	0.264 **	0.245 **	0.445 **	1

Note. ** *p* < 0.01 (2-tailed). GAS-Total Generative Acts Scale, GAS-Domestic Domestic generative acts subscale, GAS-Agentic Agentic generative acts subscale, GAS-Communal Communal generative acts subscale, GAS-Civic Civic generative acts subscale. *M* mean, *SD* standard deviation.

## Data Availability

The datasets generated during and/or analyzed during the current study are not publicly available due to datasets containing information that could compromise the privacy of research participants. The data that support the findings of this study are available from the corresponding author (H.G.) upon reasonable request.

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
