# Peer review of "Validation of the Generative Acts Scale-Chinese Version (GAS-C) among Middle-Aged and Older Adults as Grandparents in Mainland China"

_ijerph, 2021, doi:10.3390/ijerph18199950_

Round 1
Reviewer 1 Report
This article is on an important and timely topic. There is great variation in subjective well-being in custodial grandparents. Generativity is a potentially an important variable to understanding this variation. This manuscript appears to be nearly ready for publication. I suggest a few minor comments/clarifications as follows:
- line 75-77 I would like to hear more about the reasoning for distinguishing concerns from actions
- line 94-103 I see that the difference between Hong Kong and mainland China is very This information section be contained in the section 1.1 for emphasis, it is a bit confusing in the specific aims section. Similarly, references in the manuscript to 'China' might be changed to 'mainland China' to remind non-Chinese readers of the importance of this distinction
- line 112-116 The added hypothesis (?) about grandparent caregiving and generativity is misleading to the central focus of this paper as scale validation. It should be deleted
- line 130-131 Grandparent coresidence is different from living nearby. It would be helpful if you reported the number of grandparents in each category.
- line 143 The end of this line appears to be missing something. Do you mean to say that participants completed the informed consent process before taking the survey?
- line 154-155 Readers will need more information about the translation of this scale into Mandarin. Was it backtranslated?
- line 163-164 This scale is referred to as PANAS
- line 281 Should the title of this section be "Reliability?'
- The results section of this paper is an excellent example of a scale validation paper
- line 430 It would be appropriate to add level of rurality to this limiation section
Reviewer 2 Report
Kindly find the attachment below.
